# The Role of Visfatin in Gastric and Esophageal Cancer: From Biomarker to Therapeutic Target

**DOI:** 10.3390/cancers17081377

**Published:** 2025-04-21

**Authors:** Adam Mylonakis, Alexandros Kozadinos, Maximos Frountzas, Emmanouil I. Kapetanakis, Irene Lidoriki, Markos Despotidis, Eva Karanikki, Tania Triantafyllou, Dimitrios Theodorou, Konstantinos G. Toutouzas, Dimitrios Schizas

**Affiliations:** 1First Department of Surgery, Laikon General Hospital, National and Kapodistrian University of Athens, 11527 Athens, Greece; adam.mylonakis@gmail.com (A.M.); alexkozadinos@gmail.com (A.K.); markosd1995@yahoo.gr (M.D.); schizasad@gmail.com (D.S.); 2First Propaedeutic Department of Surgery, Hippocration General Hospital, National and Kapodistrian University of Athens, 11527 Athens, Greece; froumax@hotmail.com (M.F.); karanikkieva@gmail.com (E.K.); t_triantafilou@yahoo.com (T.T.); dimitheod@netscape.net (D.T.); tousur@hotmail.com (K.G.T.); 3Department of Thoracic Surgery, Attikon University Hospital, National and Kapodistrian University of Athens, 12462 Athens, Greece; 4Department of Environmental, Occupational Medicine and Epidemiology, Harvard T.H. Chan School of Public Health, Boston, MA 02115, USA; elidoriki@hsph.harvard.edu

**Keywords:** gastric cancer, esophageal cancer, biomarkers, therapeutic targets, upper gastrointestinal cancer

## Abstract

Visfatin, also known as nicotinamide phosphoribosyltransferase (NAMPT), plays a crucial role in metabolism and has been implicated in cancer progression. In gastric and esophageal cancers, visfatin is associated with tumor growth, inflammation, angiogenesis, and chemoresistance. Elevated visfatin levels correlate with poor prognosis, yet its potential as a biomarker and therapeutic target remains under investigation. This review examines the biological functions of visfatin, its role in upper gastrointestinal cancers, and its relevance as a prognostic marker. Additionally, we discuss current therapeutic strategies targeting visfatin and outline future research directions aimed at improving clinical outcomes.

## 1. Introduction

Gastric and esophageal cancers are among the most common and lethal malignancies worldwide. In 2022, gastric cancer accounted for roughly 1 million new cases (4.9% of all cancers) and was the fifth leading cause of cancer mortality (about 6.8% of cancer deaths). Esophageal cancer, while less frequent (estimated ~511,000 new cases in 2022), has a very high-case fatality rate with ~445,000 deaths globally that same year [1]. This high mortality highlights the need for better biomarkers and therapeutic targets in these malignancies.

Nicotinamide phosphoribosyltransferase (NAMPT), also identified as visfatin or a pre-B cell colony-enhancing factor (PBEF), has a complex and evolving history reflecting its multifunctionality [2]. NAMPT was first cloned and characterized as a cytokine promoting B cell precursor maturation. Subsequently, the molecule garnered attention when researchers revealed its insulin-mimetic properties, initially naming it “visfatin” due to its predominant secretion from visceral adipose tissue. Their research demonstrated that visfatin could reduce blood glucose by enhancing glucose uptake in peripheral tissues and inhibiting gluconeogenesis. However, controversy surrounding its insulin-like action led to a retraction of the initial report, as subsequent studies failed to reproduce those findings [3].

Elevated visfatin levels have been observed in metabolic disorders such as obesity and type 2 diabetes—conditions that are risk factors for gastric and esophageal malignancies. This has spurred interest in visfatin’s role in cancer development and progression, particularly in the context of obesity-associated cancers. Indeed, visfatin is often overexpressed in tumor tissues and the circulation of cancer patients, and mounting evidence links it to tumor growth, inflammation, angiogenesis, and poor prognosis [4,5,6,7,8].

Despite accumulating evidence, significant research gaps remain. There is limited understanding of the precise molecular mechanisms through which visfatin contributes to tumor progression in gastric and esophageal cancers. Additionally, the prognostic value of visfatin in clinical practice remains unclear due to variability among studies and the absence of standardized measurement methods. Furthermore, the potential therapeutic targeting of visfatin in these malignancies is underexplored, and no visfatin-targeted therapies are currently available or undergoing clinical trials.

This review aims to synthesize the current knowledge on visfatin in upper GI cancers, identify key knowledge gaps, guide future research efforts, and pave the way for more effective clinical applications.

## 2. Visfatin: Biochemistry and General Functions

Nicotinamide phosphoribosyltransferase (NAMPT) has undergone a complex evolution in its functional characterization. Initially identified as a pre-B cell colony-enhancing factor (PBEF) in 1994, NAMPT was thought to act as a cytokine involved in early B cell development [2]. Subsequent research revealed its enzymatic role in the nicotinamide adenine dinucleotide (NAD) salvage pathway, catalyzing the conversion of nicotinamide (NAM) to nicotinamide mononucleotide (NMN), a critical step in replenishing NAD pools necessary for redox reactions, DNA repair, and the regulation of sirtuins and other NAD-dependent enzymes [9,10,11,12,13].

Notably, in 2005, it was rebranded as visfatin, a visceral fat-derived hormone with proposed insulin-mimetic properties. However, this hormonal function was later contested, as subsequent studies failed to reproduce these findings [14,15,16,17]. These findings led to the retraction of the original visfatin study [3].

Intracellular NAMPT catalyzes the conversion of NAM to NMN, a critical step in the NAD salvage pathway, which replenishes the NAD pool necessary for redox reactions, DNA repair, and the regulation of sirtuins and other NAD-dependent enzymes [11,12,13].

Although the intracellular form of NAMPT (iNAMPT) plays a well-established role in regulating NAD metabolism, the biological functions of eNAMPT extend beyond its enzymatic activity. Originally thought to be secreted preferentially by visceral adipose tissue in obese patients, it is now clear that eNAMPT is produced not only by pre-B cells and adipocytes, but it is also detectable in conditioned media from various cell types [18]. A broad range of stimuli modulate eNAMPT secretion both in vitro and in vivo, including cellular stress, nutritional cues, and inflammatory signals [19]. This regulated secretion suggests a variety of specific physiological roles of visfatin under distinct conditions.

Importantly, NAMPT functions both intracellularly and extracellularly as a potent NAD biosynthetic enzyme, converting NAM to NMN. Intriguingly, eNAMPT exhibits approximately two-fold higher enzymatic activity compared to its intracellular counterpart (kcat = 0.380/s vs. 0.182/s, respectively). This enhanced extracellular enzymatic activity highlights a potentially critical systemic role for eNAMPT-derived NMN [20]. On the other hand, the NAMPT substrates are present at low concentrations or are virtually absent from the extracellular space under physiological conditions, implying that eNAMPT probably lacks sufficient substrates required for sustained and functionally relevant enzymatic activity outside the cells [21]. However, the tumor microenvironment is characterized by hypoxic areas with elevated rates of necrosis and an acidic pH and could thus bear increased levels of eNAMPT substrates and subsequently significant clinical relevance [22,23].

Beyond its role in metabolism, eNAMPT displays cytokine-like properties, significantly influencing inflammatory and immune responses. Visfatin activates antigen-presenting cells, such as monocytes, dendritic cells, and macrophages, enhancing lymphocyte activation through the upregulation of costimulatory molecules (CD80, CD40). It also modulates pro-inflammatory cytokine production, including IL-6, IL-10, and TNF-α, and acts as a potent chemotactic factor for CD14+ monocytes and CD19+ B cells, facilitating their recruitment to inflammatory sites. Consequently, visfatin has been associated with inflammatory disorders such as inflammatory bowel disease and obesity-related systemic inflammation, which could contribute to immune dysregulation and insulin resistance [24,25,26].

Additionally, visfatin promotes angiogenesis and vascular function through the upregulation of vascular endothelial growth factors (VEGFs) and matrix metalloproteinases (MMPs), mediated by the activation of mitogen-activated protein kinase (MAPK) and PI3K/Akt pathways in endothelial cells. This pro-angiogenic property positions visfatin as a crucial player in cardiovascular diseases and tumor progression, where enhanced angiogenesis is central to disease pathology [27,28,29].

## 3. The Role of Visfatin in Metabolic Disorders

Visfatin is an adipokine primarily secreted by visceral adipose tissue. It plays a crucial role in NAD biosynthesis and has been implicated in various metabolic disorders, including obesity, type 2 diabetes mellitus (T2DM), metabolic syndrome (MetS), and nonalcoholic fatty liver disease (NAFLD) [12]. However, despite numerous studies supporting its role in metabolic dysfunction, conflicting evidence challenges its exact contribution to these conditions.

Visfatin levels have been shown to be elevated in individuals with obesity, particularly in visceral adipose tissue, and correlate with inflammatory markers such as IL-6 and TNF-α [30,31]. Some studies suggest that visfatin plays a pathogenic role in obesity by promoting insulin resistance and inflammation [32]. However, other investigations question whether visfatin is merely a compensatory response rather than a direct contributor to obesity-related metabolic dysfunction. For instance, some reports indicate that visfatin levels do not significantly differ between obese and non-obese individuals [14,33]. Furthermore, weight loss interventions have yielded inconsistent effects on circulating visfatin levels, with some studies showing a decrease following weight reduction [34], while others report no significant changes [35].

Visfatin has been proposed as a potential link between obesity and insulin resistance, given its role in NAD biosynthesis and pancreatic β-cell function [20]. Elevated visfatin levels have been observed in individuals with T2DM, independent of body mass index (BMI), and have been associated with insulin resistance [36]. Some studies suggest that visfatin promotes glucose-stimulated insulin secretion, potentially compensating for β-cell dysfunction in diabetes [20]. Conversely, conflicting findings indicate that visfatin may not play a significant role in insulin resistance or glucose metabolism. While some studies report higher visfatin levels in T2DM patients, others have found no significant association between circulating visfatin and insulin sensitivity [14,15,16,17]. Additionally, experimental evidence suggests that visfatin’s insulin-mimetic effects may be overstated, as its impact on glucose metabolism appears to be mediated primarily through NAD biosynthesis rather than direct insulin receptor activation [11].

Visfatin has been proposed as a potential biomarker for metabolic syndrome (MetS), given its association with obesity, insulin resistance, and dyslipidemia [37]. Some studies report elevated visfatin levels in individuals with MetS and suggest that it may contribute to endothelial dysfunction and chronic inflammation [38,39]. Nevertheless, conflicting evidence questions whether visfatin is a reliable marker for MetS. While some studies show strong correlations between visfatin and metabolic risk factors [31], others find no significant relationship [40].

Lastly, the involvement of visfatin in NAFLD remains debated. Some studies indicate that hepatic visfatin expression is upregulated in NAFLD and correlates with liver inflammation and fibrosis [30,41]. Increased visfatin levels have also been linked to hepatic triglyceride accumulation and metabolic dysregulation [42]. However, other studies suggest an inverse relationship between visfatin and NAFLD severity. Dahl et al. reported decreased visfatin expression in patients with NAFLD, particularly in cases of advanced fibrosis [36]. Similarly, Gaddipati et al. found reduced visfatin levels in visceral adipose tissue of NAFLD patients [43].

## 4. The Role of Visfatin in Upper GI Cancer

Visfatin has emerged as a significant player in the pathophysiology of upper gastrointestinal cancers, including esophageal and gastric cancer. As a regulator of NAD metabolism and an adipocytokine, visfatin influences various oncogenic processes, including angiogenesis, tumor microenvironment modulation, cell proliferation, chemoresistance, and epithelial-to-mesenchymal transition. The following sections outline the multifaceted role of visfatin in upper GI cancer pathogenesis and progression.

### 4.1. Angiogenesis and Tumor Progression

Visfatin has been implicated in promoting angiogenesis, a critical process for tumor growth and metastasis. Huang et al. observed that visfatin levels were significantly higher in esophageal squamous cell carcinoma (ESCC) tissues compared to normal tissues. VEGF-C expression was 1.5–2 times higher in ESCC tissues than in controls. In advanced ESCC stages (IIB and IVA), visfatin levels were elevated approximately 2–3 times compared to early stages (IB and IIA). In vitro, visfatin treatment significantly increased VEGF-C mRNA and protein expression in ESCC cells in a dose-dependent manner. The inhibition of mitogen-activated protein kinase 1/2-extracellular signal-regulated kinase (MEK1/2-ERK) and the Nuclear factor kappa-light-chain enhancer of activated B cells (NF-κB) using specific inhibitors or siRNAs reduced visfatin-induced VEGF-C expression by about 50%. The authors reported that visfatin upregulates VEGF-C expression by activating the MEK1/2-ERK and NF-κB signaling pathways, leading to increased lymphovascular invasion and contributing to ESCC progression and metastasis [44].

### 4.2. Tumor Microenvironment and Immune Modulation

The tumor microenvironment (TME) plays a crucial role in cancer progression and response to therapy. Okuda et al. used single-cell RNA sequencing to investigate how neoadjuvant chemotherapy affects the tumor microenvironment in human esophageal squamous cell carcinoma (ESCC) tissue. They observed that NAMPT expression was significantly lower in various macrophage clusters in the neoadjuvant chemotherapy-treated group compared to the untreated group. They correlated this lower NAMPT expression with the reduced expression of CD36, NEAT1, PRDM1, and PTGS2; all genes associated with tumor-promoting activity and macrophage-driven immunosuppression. Thus, they suggested a shift in macrophage function in the neoadjuvant-treated group away from tumor support toward an antitumor phenotype, contributing to an enhanced antitumor immune activity in the tumor microenvironment [45].

Bi et al. studied gastric tumor tissues and gastric cancer cell lines, finding significant overexpressions of visfatin in both. They demonstrated that NAMPT mRNA levels were notably higher in cancer tissues than in adjacent normal tissues, and protein levels were similarly elevated in three gastric cancer cell lines compared to a non-cancer cell line. Furthermore, visfatin upregulated MMP2 and MMP9 expression by approximately 1.5–2-fold through activation of the NF-κB pathway [46].

In their study, Liu et al. investigated the role of visfatin, in regulating extracellular adenosine levels and its broader impact on gastric cancer progression and immune suppression. They observed that NAMPT expression was significantly elevated in gastric cancer tissues and cell lines compared to normal or adjacent non-cancerous tissues. This upregulation of NAMPT was found to play a critical role in creating an immunosuppressive tumor microenvironment. To explore this further, they conducted co-culture experiments using gastric cancer organoids and peripheral blood mononuclear cells. These experiments revealed that silencing NAMPT in the gastric cancer organoids mitigated the immunosuppressive effects exerted on CD8+ T cells, which are crucial for antitumor immunity. Specifically, when NAMPT expression was suppressed using short hairpin RNA (shNAMPT), there was a notable decrease in the expression of programmed cell death protein 1 (PD-1) on CD8+ T cells, with PD-1 expression reduced by 40% compared to controls [47].

### 4.3. Cell Proliferation and Oncogenic Signaling

Mohammadi et al. investigated the effects of visfatin on the proliferation of gastric adenocarcinoma cells and found that it increased cell proliferation by approximately 20% after 24 h. When visfatin was co-stimulated with resistin, cell proliferation increased by 30%, demonstrating a synergistic effect compared to the individual effects of either visfatin or resistin. This enhanced proliferation was mediated through the upregulation of the telomerase gene (hTERT), a gene critical for the immortality and sustained proliferation of cancer cells. Specifically, co-stimulation with both resistin and visfatin resulted in a 2.15-fold increase in hTERT gene expression, significantly higher than the 1.2-fold increase observed with visfatin alone [48].

### 4.4. Epithelial-to-Mesenchymal Transition (EMT)

EMT is a crucial biological process in tumor progression, associated with aggressive behavior, metastasis, therapeutic resistance, and poor clinical outcomes. Cancer cells undergoing EMT frequently exhibit alterations in NAD biosynthesis pathways, particularly involving the enzymes NAMPT and NAPRT. While normal cells can rely on multiple NAD biosynthetic pathways, cancer cells that have transitioned to a mesenchymal phenotype often lose NAPRT expression, typically via promoter hypermethylation or aberrant transcription initiation, thereby becoming critically dependent on NAMPT activity for survival. Additionally, elevated NAMPT expression itself promotes EMT by upregulating key transcription factors and markers, including TWIST1, VIM, and SNAI1 [49,50].

Lee et al. demonstrated this vulnerability by showing that gastric cancer cells with EMT phenotype exhibit selective sensitivity to the NAMPT inhibitor FK866. In these EMT-subtype gastric cancer cell lines, the suppression or loss of NAPRT created a synthetic lethal interaction upon the inhibition of NAMPT, as the cells lacked alternative pathways to produce NAD [50].

In their paper, Cao et al. studied in vitro gastric cancer cell lines treated with varying concentrations of recombinant visfatin to assess its effects on cell behavior. They observed that visfatin is overexpressed in gastric cancer cells by approximately 2-fold at the protein level and 1.5-fold at the mRNA level compared to normal gastric cells. Visfatin upregulated Snai1 by approximately 1.5-fold through the activation of the NF-κB signaling pathway, contributing to the epithelial–mesenchymal transition (EMT) process and facilitating the migration and invasion of gastric cancer cells [51].

An overview of the suggested and established roles of intracellular and extracellular NAMPT in upper gastrointestinal cancer pathogenesis is presented in Figure 1.

## 5. The Application of Visfatin as a Tumor Biomarker in Upper GI Cancer

The role of NAMPT as a tumor biomarker is still under investigation, with limited and conflicting data available.

Lu et al., in a clinical study involving 262 gastric cancer patients and 262 matched healthy controls, investigated the role of eNAMPT as a biomarker. Plasma eNAMPT levels were significantly elevated in gastric cancer patients compared to healthy individuals (mean cutoff value: 78.4 ng/mL, *p* < 0.001). Elevated plasma eNAMPT was strongly correlated with adverse clinicopathological features, including increased tumor invasion depth, lymph node metastasis, distant metastasis, peritoneal dissemination, larger tumor size (≥5 cm), and advanced TNM stage (*p* < 0.005 for all). Multivariate analysis identified high plasma eNAMPT levels as an independent predictor of increased 5-year mortality (OR: 3.829; 95% CI: 1.838–7.308; *p* = 0.004) and reduced overall survival (HR: 2.973; 95% CI: 1.871–5.201; *p* = 0.006). Additionally, receiver operating characteristic curve analysis confirmed plasma eNAMPT’s strong predictive capability for mortality outcomes, highlighting its potential as a valuable prognostic biomarker for gastric cancer [52].

Takahashi et al. identified NAMPT as a significant biomarker in perioperative esophageal cancer patients managed in the ICU. NAMPT mRNA levels increased notably from postoperative day (POD) 5 onward, with elevated NAMPT mRNA expression on POD 3 independently associated with increased 1-year mortality (multivariate regression, *p* = 0.007). Among the twenty-seven patients studied, four (14.8%) with severe complications (such as sepsis following anastomotic leakage or aspiration pneumonia) showed significant NAMPT upregulation (*p* < 0.001 and *p* = 0.025, respectively). Furthermore, treatment with sivelestat, a neutrophil elastase inhibitor, was associated with a reduction in NAMPT mRNA levels (*p* = 0.045). While the study did not establish a direct causal relationship, this finding suggests a potential link between sivelestat administration and the modulation of NAMPT-associated inflammation [53].

Liu et al., using a combination of in vivo mouse models and in vitro cell culture experiments, investigated how obesity influences the growth and progression of esophageal squamous cell carcinoma in relation to visfatin. The authors demonstrated that serum levels of visfatin were markedly higher in obese mice, and a strong positive correlation was observed between visfatin levels and tumor weight. Additionally, ESCC cells cultured with serum from obese mice showed increased proliferation, migration, and invasion, alongside the upregulation of pro-tumorigenic markers, such as MMP9 and YAP, and the downregulation of the energy-sensing protein AMPK. These findings suggested that obesity promotes ESCC progression through endocrine mechanisms, with visfatin potentially acting as a mediator by modulating the AMPK-YAP axis, thus contributing to cancer cell survival and tumor progression [54].

Nakajima et al. analyzed serum visfatin levels in 156 gastric cancer patients and 156 controls. Their results showed that visfatin levels were significantly higher in gastric cancer patients compared to controls and that these levels increased progressively with advancing tumor stage. Interestingly, visfatin levels did not show a clear correlation with BMI in either patients or controls, suggesting that visfatin may serve as an independent biomarker for gastric cancer progression, irrespective of BMI [55].

On the contrary, the same author team reported that serum visfatin levels were not significantly associated with ESCC. Specifically, visfatin concentrations showed no meaningful difference between ESCC patients and control subjects, nor did they correlate with BMI. Thus, visfatin was not identified as a potential biomarker for ESCC in this study [56].

The study conducted by Liu et al. investigated the role of NAMPT as a biomarker in esophageal squamous cell carcinoma, particularly in relation to obesity, chemotherapy response, and prognosis. Serum eNAMPT levels were significantly elevated in overweight and obese ESCC patients (8.44 ± 0.83 vs. 6.82 ± 1.68 ng/mL, *p* = 0.001) and correlated with markers of adiposity, including body mass index, waist circumference, and total fat area (r = 0.50, *p* < 0.001). However, eNAMPT was not independently associated with chemotherapy response or survival outcomes. In contrast, iNAMPT expression was significantly upregulated in ESCC tumor tissues compared to adjacent non-tumor tissues at both the mRNA (1.9 ± 0.3 vs. 1.0 ± 0.2, *p* < 0.05) and protein levels (5.73 ± 1.03% vs. 2.12 ± 0.67%, *p* < 0.05). Elevated iNAMPT was an independent predictor of poor response to neoadjuvant chemotherapy (OR = 2.00, *p* = 0.004) and was associated with significantly reduced disease-specific survival (median survival: 36.82 ± 2.48 vs. 57.71 ± 3.92 months, HR = 3.50, *p* < 0.001). Additionally, tumor regression grading scores of 2–3 were independently associated with worse survival outcomes (HR = 3.84, *p* < 0.001), while lymph node metastasis post-therapy was another independent prognostic factor (HR = 3.79, *p* < 0.001) [57].

Li et al. investigated the role of NAMPT in esophagogastric junction adenocarcinoma, researchers found that serum eNAMPT levels were significantly higher in overweight and obese patients, particularly those with visceral obesity, and these elevated levels positively correlated with body mass index, waist circumference, visceral fat area, as well as tumor progression indicators such as primary tumor size, regional lymph node involvement, and TNM stage; additionally, iNAMPT expression was upregulated in esophagogastric junction tissues and associated with worse survival outcomes, suggesting that both eNAMPT and iNAMPT are linked to visceral obesity and may influence tumor biology [58]. 

In summary, current evidence suggests that eNAMPT holds potential as a biomarker in gastric cancer, with significant correlations to tumor progression and survival. However, its role in esophageal cancer remains uncertain, as findings indicate stronger associations with obesity rather than tumor biology. iNAMPT, in contrast, appears to be a more robust prognostic marker in ESCC, particularly for chemotherapy response and survival outcomes.

## 6. The Inhibition of Visfatin as Therapeutic Approach for Upper GI Cancer

Given its role in NAD metabolism, inflammation, and tumor survival, NAMPT inhibition represents a promising therapeutic strategy. Targeting iNAMPT disrupts cancer cell energy metabolism, while an eNAMPT blockade may suppress tumor-promoting inflammation and angiogenesis.

Intracellular NAMPT is primarily targeted using small molecule inhibitors, such as FK866, CHS828, and OT-82. By inhibiting NAMPT, intracellular NAD levels are depleted, leading to impaired energy metabolism and subsequent cancer cell death [59,60]. Additionally, these agents can exert their anticancer effects through the inhibition of NF-κB activity. This inhibition is thought to occur via the suppression of IκB kinase beta (IKKβ), leading to reduced NF-κB signaling, which is often associated with cancer cell survival and proliferation [61,62]. Notably, the inhibition of NAMPT using FK866, a specific non-competitive inhibitor, depletes NAD levels, leading to suppressed cell growth, migration, and anchorage-independent growth. FK866 significantly downregulates VEGF, MMP2, MMP9, and NF-κB, key factors in tumor progression. At concentrations of 3–30 nM, FK866 reduced gastric cancer cell proliferation after 48–72 h (*p* < 0.05), and, at 30 nM, it markedly inhibited cell migration. Additionally, FK866 sensitized gastric cancer cells to fluorouracil (5-FU), significantly lowering the half-maximal inhibitory concentration (IC50) and enhancing apoptosis when combined with 5-FU [46].

Interestingly, a study conducted by Lee et al. demonstrated that FK866, a NAMPT inhibitor, selectively kills gastric cancer cells with an epithelial–mesenchymal transition gene signature due to their deficiency in NAPRT, which makes them reliant on NAMPT for NAD synthesis. In vitro and in vivo experiments showed that FK866 induced significant cytotoxicity in EMT-subtype gastric cancer cell lines and tumors but spared non-EMT cells, with tumor regression observed in xenograft models. The loss of NAPRT, frequently driven by promoter hypermethylation, was correlated with EMT markers and stabilization of β-catenin, suggesting a role in tumor progression. Importantly, NAPRT knockdown sensitized resistant cells to FK866, while its overexpression protected EMT cells from toxicity, highlighting NAPRT as a predictive biomarker for NAMPT inhibitor therapy. These findings suggest that FK866 could be a promising targeted therapy for EMT-subtype gastric cancers, which are often aggressive and drug-resistant [50].

However, clinical trials with NAMPT inhibitors of this type have demonstrated modest efficacy, primarily due to dose-limiting toxicities such as thrombocytopenia, gastrointestinal effects, retinal, neurological, and cardiac toxicities [63]. Thus, optimizing pharmacokinetics or combining with other agents remains necessary to improve therapeutic outcomes.

Recent innovative therapeutic strategies involve dual-targeting molecules. For instance, the inhibition of NAMPT using agents like GMX1778 depletes cellular NAD levels, disrupting CtBP dimerization and its oncogenic transcriptional functions. This dual inhibition strategy significantly reduces the transcription of oncogenic genes, such as TIAM1, leading to decreased cancer cell proliferation, migration, and tumor growth in xenograft models. Importantly, this combination therapy, when used in pancreatic cancer cells, has demonstrated potent efficacy in blocking pancreatic cancer cell growth both in vitro and in vivo with no observable toxicity in preclinical studies [64]. Another dual inhibitor, STF-31, targets NAMPT and GLUT1, exerting potent antitumor activity by interfering with both NAD salvage pathways and glucose metabolism [65].

To further reduce the reported toxicity of small-molecule inhibitors, researchers are developing new novel drug delivery strategies for NAMPT inhibitors. These include antibody–drug conjugates (ADCs) that deliver NAMPT inhibitors specifically to cancer cells, as well as proteolysis-targeting chimera (PROTAC) technology NAMPT degraders that target both intracellular and extracellular forms of the enzyme [66,67].

Extracellular NAMPT, functioning independently of its enzymatic NAD synthesis role, acts as an cytokine that promotes inflammation and tumor progression through pathways involving MAPK, NF-κB, and PI3K/Akt signaling cascades. Extracellular NAMPT targeting can be effectively approached using mainly novel drug delivery approaches including PROTACs and anti-eNAMPT neutralizing antibodies, which selectively degrade extracellular NAMPT [68,69].

Regarding visfatin receptor targeting, though initially unclear, recent evidence suggests that its pro-oncogenic actions may be mediated by undefined receptors or indirectly via signaling pathways such as PI3K/Akt and MAPK. Thus, identifying and targeting these receptors or downstream pathways in the future could enhance therapeutic specificity.

A comprehensive summary of the therapeutic strategies targeting NAMPT in upper GI cancer is presented in Table 1. In summary, despite promising preclinical data, NAMPT inhibitors face challenges in clinical translation due to toxicity. Future strategies should focus on optimizing combination regimens and advanced drug delivery systems to enhance selectivity and minimize adverse effects.

## 7. Future Directions and Research Needs

### 7.1. What Is Missing About Visfatin’s Role in Upper GI Cancers?

Despite increasing the recognition of visfatin’s role in upper GI cancers, several key research gaps remain. First, the molecular mechanisms by which visfatin drives tumor progression and therapy resistance remain incompletely understood. In particular, the interaction between intracellular and extracellular NAMPT is poorly defined. It is not yet known whether they work synergistically to promote tumor progression, perform similar roles, and can compensate for each other or have different effects depending on the tumor environment or disease stage. Elucidating these interactions and mechanisms will be vital to guide targeted therapeutic development.

Additionally, the optimal source for visfatin measurement remains unclear. While visfatin is mainly secreted by visceral fat, its levels can be detected in blood, adipose tissue, and tumor biopsies [70]. However, whether serum visfatin, tumor-associated visfatin, or adipose-derived visfatin is most predictive of cancer progression has not been established. Standardizing measurement techniques is crucial to improving its reliability as a biomarker and enabling its use in clinical decision making [52].

Furthermore, the relationship between visfatin levels and surgical outcomes has not been well defined. While visfatin is linked to tumor aggressiveness, its potential impact on post-surgical recovery, immune response, and recurrence risk has not been systematically studied. Understanding fluctuations in visfatin levels before and after surgery could provide insights into its role in wound healing, inflammation, and tumor recurrence dynamics. Furthermore, whether postoperative visfatin levels correlate with disease-free survival or metastasis risk remains unknown.

Another unexplored area is visfatin’s potential as a dynamic marker of treatment response. The longitudinal monitoring of visfatin levels during chemotherapy could provide a real-time assessment of treatment efficacy and resistance development. While some evidence suggests that high visfatin levels correlate with chemotherapy resistance, no studies have evaluated whether declining visfatin levels predict better therapeutic response or improved prognosis. Prospective studies tracking visfatin levels before, during, and after chemotherapy could help determine whether it serves as a reliable biomarker for treatment monitoring.

Furthermore, the role of visfatin in chemotherapy resistance remains insufficiently characterized. Visfatin has been implicated in resistance mechanisms in various cancers, yet its specific contribution to drug resistance in gastric and esophageal cancers is not fully understood [71,72]. Investigating how visfatin interacts with drug efflux transporters, DNA repair pathways, and survival signaling cascades (such as PI3K/Akt and NF-κB) could reveal new therapeutic targets. Additionally, preclinical studies are needed to assess whether targeting visfatin with inhibitors such as FK866 enhances chemotherapy sensitivity in resistant tumors.

The link between visfatin, metabolic dysfunction, and cancer progression is another critical area requiring further investigation. Diabetes and obesity—both conditions related to elevated visfatin levels—are known risk factors for gastric and esophageal cancers [73]. However, the direct mechanistic relationship between hyperinsulinemia, visfatin upregulation, and tumorigenesis remains unclear. Future studies should evaluate whether targeting visfatin in obese or diabetic patients with upper GI cancers could improve outcomes and whether metabolic interventions influence visfatin’s oncogenic effects.

Finally, large-scale clinical studies are needed to validate visfatin as a prognostic biomarker. While small studies suggest that high visfatin levels correlate with tumor progression and poor survival, more extensive trials are required to establish its independent predictive value in upper GI cancers. If validated, visfatin could assist in risk stratification, guiding personalized treatment approaches, and identifying high-risk patients who may benefit from more aggressive interventions.

Addressing these gaps will be essential for determining visfatin’s clinical relevance and potential as a therapeutic target. Future research should focus on standardizing measurement techniques, clarifying its role in therapy resistance and metabolic dysfunction, and assessing its utility in clinical practice.

### 7.2. State-of-the-Art Assessment of Visfatin

Recent advances in analytical techniques have significantly improved the precision and reliability of visfatin assessments in both clinical and research settings. However, standardization remains a critical challenge, as differences in detection methods contribute to variability in study results.

The most commonly used method for visfatin quantification is the enzyme-linked immunosorbent assay (ELISA), which allows for the rapid and sensitive detection of visfatin levels in blood and tissue samples. While ELISA-based methods provide reproducible results, they can be influenced by batch-to-batch variability, cross-reactivity with other proteins, and differences in antibody specificity. Mass spectrometry-based approaches have emerged as a more precise alternative, offering higher sensitivity and specificity, particularly in studies evaluating visfatin’s role in metabolic diseases and malignancies. However, mass spectrometry is not widely available in clinical settings due to its cost and technical complexity.

Another promising approach is the measurement of visfatin in alternative biofluids, such as saliva. Salivary visfatin has been studied in the context of gestational diabetes mellitus, demonstrating its potential as a non-invasive biomarker [74]. However, its application in cancer diagnostics remains unexplored. Future studies should evaluate whether salivary visfatin correlates with systemic or tumor-specific visfatin levels in upper GI cancers, as this could provide a less invasive method for monitoring disease progression and treatment response.

Additionally, adipose tissue biopsy could offer a more localized and functionally relevant measure of visfatin levels, particularly in patients with obesity-associated cancers. Given that visfatin is primarily secreted by visceral fat, sampling specific fat depots (e.g., visceral versus subcutaneous fat) may provide greater insights into its metabolic and oncogenic roles. However, the clinical feasibility and utility of routine adipose tissue biopsy remain uncertain, particularly in patients undergoing cancer treatment.

While imaging technologies such as PET scans have revolutionized cancer diagnostics, there are currently no established imaging modalities to track visfatin activity or distribution in vivo. Future research should explore whether radiolabeled visfatin-specific probes could enable the visualization of visfatin-expressing tumors, providing real-time metabolic and oncogenic insights.

In summary, standardized, sensitive, and clinically accessible methods for visfatin measurement are needed to establish its role as a reliable biomarker in upper GI cancers. Advances in high-sensitivity assays, non-invasive biofluid analysis, and emerging imaging technologies could enhance our understanding of visfatin’s function and improve its clinical applicability.

### 7.3. Proposals for Future Research Focusing on Translational and Clinical Studies

Despite growing evidence supporting visfatin’s role in upper gastrointestinal (GI) malignancies, significant gaps remain in translating these findings into clinical applications. Future research should focus on mechanistic studies, biomarker validation, and targeted therapeutic approaches, ultimately integrating visfatin-based strategies into personalized oncology.

To fully elucidate visfatin’s oncogenic mechanisms, future studies should investigate its role in tumor microenvironment modulation, immune evasion, and metabolic reprogramming. A key area of interest is visfatin’s interaction with other adipokines (e.g., leptin, resistin, chemerin) and their combined effects on tumor progression. Additionally, given its influence on inflammatory signaling (e.g., NF-κB, PI3K/Akt, MAPK), studies should explore how visfatin modulates immune cell infiltration and therapy resistance in esophageal and gastric cancer.

Preclinical studies using genetically modified models (e.g., visfatin knockout mice or CRISPR-engineered cell lines) can provide definitive evidence of its role in tumorigenesis. Additionally, the impact of visfatin on EMT, angiogenesis, and metastatic spread should be evaluated using organoid models and patient-derived xenografts, which better mimic human disease conditions.

To establish visfatin as a clinically useful biomarker, large-scale, multicenter clinical trials should evaluate its prognostic and predictive value. Future studies should aim to

Determine optimal cutoff values for visfatin levels in serum, tissue, and alternative biofluids (e.g., saliva, peritoneal fluid).Assess whether preoperative visfatin levels predict surgical outcomes, recurrence risk, or complications.Investigate whether visfatin dynamics during chemotherapy or immunotherapy correlate with treatment response and survival.Compare visfatin levels across different tumor subtypes (intestinal vs. diffuse gastric cancer, esophageal adenocarcinoma vs. squamous cell carcinoma) to identify disease-specific patterns.

Given the conflicting findings regarding visfatin’s prognostic significance, studies should also stratify patients by metabolic status (obesity, diabetes, metabolic syndrome) to determine whether visfatin’s role is context-dependent.

The successful translation of visfatin research into clinical oncology will require close collaboration between basic scientists, oncologists, and pharmaceutical researchers. Establishing data-sharing initiatives to pool biomarker data from diverse cancer cohorts will be essential for standardizing visfatin as a prognostic and predictive marker. Additionally, efforts should focus on identifying potential therapeutic applications by repurposing existing drugs that modulate visfatin pathways, which could accelerate clinical implementation. Collaborative studies integrating clinical, molecular, and histopathological data will provide a more comprehensive understanding of visfatin’s role in tumor progression and treatment response, ultimately paving the way for more targeted and effective therapies in upper gastrointestinal cancers.

## 8. Conclusions

Visfatin, a multifunctional cytokine and adipokine, plays a pivotal role in both inflammatory and metabolic processes, positioning it as a promising prognostic biomarker and therapeutic target in upper gastrointestinal cancers. Accumulating evidence highlights visfatin’s involvement in upper gastrointestinal cancers, with studies demonstrating its role in tumor progression, immune modulation, chemoresistance, and metabolic dysfunction. While elevated visfatin levels have been associated with poor prognosis and aggressive tumor behavior, inconsistencies across studies underscore the need for standardized assessment methods and large-scale clinical validation. The potential of visfatin as both a biomarker and therapeutic target remains under investigation, with ongoing research exploring its role in metabolic pathways and tumor microenvironment interactions. Collaborative efforts integrating clinical, molecular, and therapeutic studies will be essential for refining its clinical utility and identifying targeted treatment strategies. Future studies should prioritize the development of visfatin-directed therapies and their integration into existing treatment protocols to enhance patient outcomes in gastric and esophageal cancers.

## Figures and Tables

**Figure 1 cancers-17-01377-f001:**
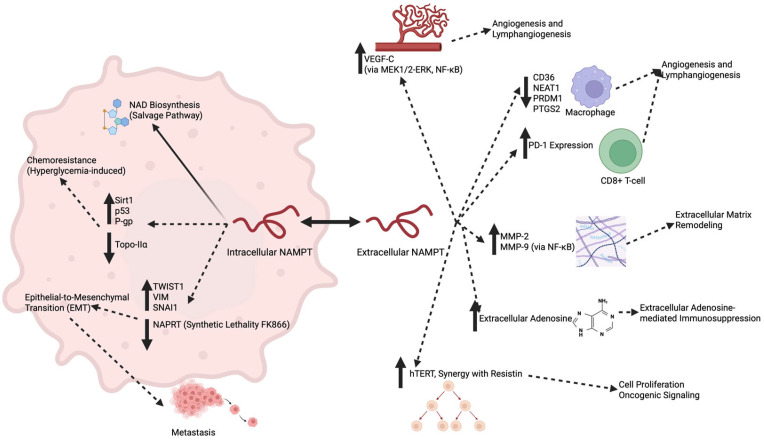
Suggested and established roles of intracellular and extracellular NAMPT in upper gastrointestinal cancer pathogenesis. Intracellular NAMPT within cancer cells maintains NAD levels (established mechanism) essential for cancer cell metabolism and survival. Suggested intracellular functions (dashed arrows) include promoting chemoresistance under hyperglycemic conditions via the altered expression of resistance-related proteins (Sirt1, P-gp, Topo-IIα) and facilitating EMT through the increased expression of transcription factors (TWIST1, VIM, SNAI1). Extracellular NAMPT, secreted into the tumor microenvironment, has several suggested roles (dashed arrows), including the promotion of angiogenesis and lymphangiogenesis (VEGF-C, MEK1/2-ERK, NF-κB signaling), the modulation of immune responses (CD36, NEAT1, PRDM1, PTGS2, PD-1 on CD8+ T cells), ECM remodeling (MMP-2, MMP-9), extracellular adenosine-mediated immunosuppression, and the stimulation of tumor cell proliferation (hTERT, resistin). Solid arrows denote proven mechanisms, while dashed arrows represent mechanisms that are currently suggested but require further validation (Created in BioRender.com on 20 March 2025).

**Table 1 cancers-17-01377-t001:** Therapeutic strategies targeting NAMPT in upper gastrointestinal cancer.

Targeted NAMPT Form	Therapeutic Agents	Mechanism of Action	Observed Antitumor Effects	Clinical/Experimental Outcomes	Limitations/Toxicity
Intracellular NAMPT (iNAMPT)	FK866, CHS828, OT-82	Small-molecule inhibitors; deplete NAD levels by inhibiting NAMPT enzymatic activity; suppress NF-κB signaling (via IKKβ inhibition)	Inhibited cancer cell proliferation, migration, anchorage-independent growth; enhanced apoptosis; sensitization to chemotherapy (5-FU); downregulation of VEGF, MMP2, MMP9, NF-κB (p65)	Reduced gastric cancer cell viability and migration at 3–30 nM FK866; significant sensitization to 5-FU (lower IC50)	Thrombocytopenia, gastrointestinal effects, retinal, neurological, cardiac toxicities observed in clinical trials
Intracellular NAMPT (iNAMPT)(EMT-SubType Specific)	FK866	Selectively induces cytotoxicity in EMT-subtype gastric cancer cells lacking NAPRT due to promoter hypermethylation	Significant in vitro and in vivo antitumor activity specifically against EMT-subtype tumors; tumor regression in xenograft models; NAPRT as a predictive biomarker	Effective selective killing of aggressive EMT-subtype gastric cancer cells; no activity against non-EMT cells	Potential limitation in tumors expressing NAPRT (resistance)
Intracellular NAMPT (iNAMPT) + CtBP Dual Inhibition	GMX1778 + 4-Cl-HIPP	Dual-targeting: NAMPT inhibition (GMX1778) disrupts CtBP dimerization; 4-Cl-HIPP inhibits CtBP oligomerization and oncogenic gene transcription	Reduced cancer cell proliferation, migration, transcription of oncogenic genes (e.g., TIAM1); effective in pancreatic cancer xenograft models	Potent in vitro and in vivo efficacy; no observable preclinical toxicity	Awaiting clinical validation
Intracellular NAMPT (iNAMPT) + GLUT1 Dual Inhibition	STF-31	Inhibits both NAMPT and GLUT1; disrupts NAD metabolism and glucose uptake	Potent antitumor activity by interfering with metabolic pathways	Significant antitumor effects in preclinical models	Awaiting clinical validation
Intracellular and Extracellular NAMPT (iNAMPT and eNAMPT)	PROTAC-based NAMPT degraders; Antibody-Drug Conjugates (ADCs)	Novel drug delivery systems: Selectively deliver NAMPT inhibitors and promote targeted degradation of NAMPT	Enhanced specificity; simultaneous targeting of intracellular and extracellular NAMPT; reduced toxicity	Preclinical effectiveness demonstrated; selective targeting achieved	Early experimental stages; clinical translation pending
Extracellular NAMPT (eNAMPT)	Anti-eNAMPT neutralizing antibodies; PROTAC technology	Neutralization or degradation of extracellular cytokine activity of eNAMPT independent of NAD synthesis; inhibition of inflammatory signaling (MAPK, NF-κB, PI3K/Akt)	Suppression of tumor-associated inflammation, angiogenesis, and tumor progression	Effective experimental reduction in eNAMPT-mediated inflammatory and pro-tumor effects	Awaiting clinical validation; precise receptors remain unclear
Visfatin Receptor Pathway (Future Perspectives)	Future receptor-specific agents or downstream inhibitors (PI3K/Akt, MAPK)	Identification and targeting of undefined visfatin receptors or downstream signaling pathways	Potential enhancement of specificity; suppression of pro-oncogenic signaling	Hypothesized improvement in therapeutic specificity and efficacy	Currently theoretical; receptor identity and efficacy need validation

Abbreviations: NAMPT, nicotinamide phosphoribosyltransferase; EMT, epithelial–mesenchymal transition; NAD, nicotinamide adenine dinucleotide; NAPRT, nicotinic acid phosphoribosyltransferase; VEGF, vascular endothelial growth factor; NF-κB, nuclear factor-kappa B; GLUT1, glucose transporter 1; CtBP, C-terminal binding protein; PROTAC, proteolysis-targeting chimera; ADCs, antibody-drug conjugates; MAPK, mitogen-activated protein kinase; PI3K, phosphatidylinositol 3-kinase; Akt, protein kinase B; 5-FU, 5-fluorouracil; and IC50, half-maximal inhibitory concentration.

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
