# Peer review of "The Role of Visfatin in Gastric and Esophageal Cancer: From Biomarker to Therapeutic Target"

_cancers, 2025, doi:10.3390/cancers17081377_

Round 1

Reviewer 1 Report

Comments and Suggestions for Authors

Kindly find my comments on the manuscript.

The authors provided a brief overview of the roles of NAMPT (also known as visfatin) as both an intracellular enzyme and an extracellular adipokine/cytokine involved in diverse cellular processes. They then chose to focus on the contribution of NAMPT/visfatin in gastric and esophageal cancers, highlighting its potential exploitation as a tumor biomarker and/or as a therapeutic target in these two specific types of upper GI tumors.
However, I suggest the following points that could further improve the quality of the manuscript.
1.
Kindly introduce the acronym only after the first mention of the full name, and then use the acronym consistently without mentioning the full name. for example, NAD was introduced in line 55 and then in line 81 (also for consistency, either use NAD or NAD+ throughout the manuscript); Vascular Endothelial Growth Factor (VEGF) and matrix metalloproteinases (MMPs) in lines 93-94 and then again in 163-164; nicotinamide mononucleotide (NMN) in lines 85 and 145; NAM was introduced as an acronym for nicotinamide in line 84 but the full word was used instead of NAM used in line 144 and the same for ESCC; NAPRT line 417 was not introduced. Kindly revise through the manuscript.
2.
Also, I would recommend substantiating your statements with references whenever necessary throughout the text. For example, in lines 77-78 “It is synthesized predominantly in the visceral fat tissue, but has been also detected in skeletal muscle, liver, cardiomyocytes, and immune cells.” Add references. Also, section 2.2.3. (Lines 150-160) can be backed up by more evidence
3.
There are a lot of repetitive parts in Section 2.1 and Section 2.2 that convey more or less the same information (for example, the catalytic function of NAMPT as a NAD-producing enzyme is highlighted in section 2.1. and then section 2.2.2, the “insulin-mimetic” role of visfatin and its implication in diabetes and obesity and in driving angiogenesis. To avoid repetition, I would suggest linking the functions or the implications mediated by NAMPT as an intracellular enzyme or as an extracellular ligand with the underlying metabolic and/or signaling pathways.
4.
The insulin-mimetic function of visfatin is a controversial aspect. Indeed, the main paper that introduced the insulin-mimetic function of NAMPT (Reference no.2, which is also not mentioned in the MS) has been retracted (PMID: 17962537). This aspect should be better discussed in the manuscript, taking into account the available studies that also don’t support this role (for example, PMID: 21327328, PMID: 21251239, PMID: 17983582). The involvement of NAMPT enzymatic activity in mediating these effects should also be considered. In addition, the claimed ability of visfatin to mimic insulin’s activity is a role/function rather than a specific pathway, so is there a certain reason that it is discussed in the section specified for the pathways triggered by visfatin (section 2.1.5)?
5.
In the introduction lines, the statements that NAMPT/VISFATIN was “primarily identified in visceral adipose tissue” and “Visfatin was initially recognized for its insulin-mimetic effects” could be a bit misleading. NAMPT was originally characterized as a pre-B-cell colony-enhancing factor (PBEF) in 1994 by Samal et al. (PMID: 8289818) Also, the enzymatic activity of NAMPT was described much earlier. Kindly correct these statements, providing a more precise description of the history and the chronological characterization of the relatively complex functionality of NAMPT (PMID: 19149599).
6.
Line 141-142 “Notably, visfatin has also been implicated in the pathophysiology of gestational diabetes mellitus by influencing glucose and lipid metabolism” & Line 157-160 “Thus, visfatin is implicated in the pathogenesis of inflammatory disorders, including inflammatory bowel disease and systemic inflammation associated with obesity, where it mediates immune dysregulation, inflammation and insulin resistance”. These are strong conclusions/statements that need to be supported by stronger evidence other than the cited articles. Kindly revise.
7.
2.2.5. Line 169-173: “Visfatin is intricately involved in lipid metabolism, particularly in the context of obesity. Elevated visfatin levels are associated with increased lipid storage in adipocytes and dysregulation of lipid profiles, which contributes to the development of metabolic disorders, especially in diabetic patients [33]. This is partly due to visfatin's ability to modulate the expression of genes
involved in lipid synthesis and storage
[3].” Again, the evidence presented in this paragraph doesn’t support the strong and definite conclusions presented by the authors. Kindly revise.
8.
Generally, this section (i.e., 2. Visfatin: Biochemistry and General Functions) could be modified in a better way, while taking into consideration avoiding redundancy, presenting solid evidence to support the authors’ statements/conclusions, and discussing in a more detailed yet comprehensive way the various functions of visfatin linking them to their mechanistic pathways if possible. In addition, since the authors particularly focus on gastric and esophageal tumors in the following sections, the pro-oncogenic roles of visfatin could also be introduced/highlighted in this section.
9.
“It is the sixth leading cause of cancer-related deaths worldwide, with a growing prevalence, particularly in developed countries [35].” This estimation is based on the Global Cancer Statistics 2018 data (PMID: 30207593). In the recently published Global Cancer Statistics 2022 data (PMID: 38572751), esophageal cancer has dropped from the sixth to the seventh leading cause of cancer-related deaths. The same comment also applies to gastric cancer (Lines 246-248). Kindly update.
10.
Ref.36 is not mentioned in the text, and ref.39 is mentioned before ref.38.
11.
“increased ESCC cell viability by approximately 25%” Line 196. Supplementary Figure 3 in the cited paper doesn’t support this conclusion. Kindly check.
12.
“Liu et al. correlated positively tumor weight with visfatin levels, using a xenograft tumor in an in-vivo animal model [39]. Visfatin was implicated in altering the Yes-associated protein (YAP) AMPK signaling pathway promoting ESCC tumor growth and progression in the context of obesity [38].” Lines 198-201. I believe you mean ref. (38) instead of (39) (The author’s name in ref. 39 is not Liu). If yes, kindly correct it. In the first sentence, kindly clarify the context/design of this in vivo study more clearly. Also, the cited reference doesn’t support the conclusion stated in the second sentence. Kindly revise.
13.
“esophagogastric adenocarcinoma tumor tissues” Lines 202-203. I believe the word junction is missing. If yes, kindly correct.
14.
One study by Nakajima and colleagues (who previously suggested that visfatin may act as a biomarker of gastric cancer; ref 52) failed to detect a significant difference in visfatin levels between patients with squamous cell carcinoma of the esophagus and controls (PMID: 19693538). This controversy could be at least briefly highlighted in the context of esophageal cancer (PMID: 27128025).
15.
In lines 221-232 discussing the findings of ref.40: Kindly revise this paragraph and provide a more accurate description of the findings related to NAMPT in this study. Kindly take into account the differences between the actual experimental results and the suggested mechanisms /implications /explanations /commentaries provided by the authors on their findings.
16.
“In their study, Lu et al. demonstrated that visfatin plays a crucial role in the progression and metastasis of gastric cancer by promoting various oncogenic processes [49]. Their research, involving 262 gastric cancer patients, showed that visfatin is overexpressed in gastric cancer tissues, with a notable 60.4% of patients with tumor sizes ≥5cm exhibiting high visfatin levels [49]. Their study highlighted that visfatin activates key signaling pathways such as PI3K/Akt and NF-κB, which are essential for tumor growth and survival. Additionally, visfatin was shown to upregulate matrix metalloproteinases (MMPs), enzymes that degrade the extracellular matrix, thus facilitating the invasion of cancer cells into adjacent tissues and promoting distant metastasis [49]. These findings suggest that visfatin significantly contributes to the aggressive nature of gastric cancer, with 34% of patients with distant metastasis exhibiting elevated visfatin levels, highlighting its potential role as a target for therapeutic intervention [49].” Lines 262-273. While the cited reference shows that visfatin levels are elevated in GC patients, correlated with tumor aggressiveness and poor prognosis and that it can serve as a prognostic tool/biomarker, it doesn’t provide experimental evidence for the described oncogenic roles or for the underlying pathways of visfatin. Kindly revise the paragraph to reflect the findings of the study more accurately.
17.
“Zhao et al. demonstrated that visfatin plays a significant role in promoting chemo resistance and progression in gastric cancer, particularly under high glucose conditions [50].The study highlighted that visfatin expression was significantly higher in gastric cancer tissues, especially in patients with
diabetes, and that this overexpression was associated with increased resistance to chemotherapy,
such as 5-fluorouracil (5-FU) [50]”. The study found that NAMPT expression was elevated in gastric cancer patients with diabetes compared to those without diabetes, and this increased NAMPT expression was associated with poorer overall survival in the diabetic GC patients. It was shown that high glucose levels can increase NAMPT expression and can mediate resistance to 5-FU in gastric cancer cells. However, the study does not directly establish a causal relationship between NAMPT overexpression and chemotherapy resistance, and therefore, a direct link between elevated NAMPT expression and increased resistance to CT can’t be concluded/stated (at least based on the results presented in the cited study). Kindly revise this paragraph to describe the findings of this study more accurately.
18.
“They observed that NΑΜPT enhances the activity of SIRT1, a NAD+-dependent deacetylase, which in turn deactivates PARP1, protecting cells from PARP1-mediated cell death [53]. “ Lines 305-307: In which section of the study “ref.53” were these findings presented? I only found a part in the introduction briefly highlighting the cross-talk between NAMPT, SIRT1, and PARP1.
19.
“They found that in 6 out of 7 gastric cell lines with an epithelial-mesenchymal transition (EMT) gene expression signature, visfatin was found to be overexpressed. [54].” This sentence could be misleading as NAMPT/visfatin expression is not the key determinant in this context, and its levels of expression were not even highlighted in the mentioned study. Instead, the study emphasizes that the characteristic feature of these cell lines is their diminished NAPRT expression that accompanies the EMT subtype and underlies their extreme sensitivity to the NAMPT inhibitor FK866. NAPRT is a NAD+ biosynthetic enzyme that, if expressed, allows tumors to circumvent NAMPT inhibition by producing NAD+ from nicotinic acid rather than nicotinamide. Kindly revise and correct this sentence to reflect the findings more accurately.
20.
“Serum visfatin levels are significantly elevated in HCC patients compared to those without HCC and healthy controls, correlating with hepatic dysfunction, advanced tumor stages, larger tumor size, and poorer overall survival [60].” Lines 366-369. The difference between those without HCC and healthy controls is not clear. In the cited paper, I believe there were three cohorts: 1) cirrhotic patients with HCC, 2) patients with liver cirrhosis without HCC, and 3) healthy controls. Also, this study doesn’t support some of the mentioned correlations. Kindly revise.
21.
“Visfatin promotes HCC cell proliferation, invasion, and lipogenesis by activating the Akt and ERK signaling pathways, increasing reactive oxygen species (ROS) production, and elevating matrix metallopeptidase 9 (MMP-9) activity, which contributes to an invasive cancer phenotype [61].” Kindly rephrase this sentence more accurately. While the study describes several pro-oncogenic effects of visfatin, the direct link between the pro-tumorigenic effects of visfatin (e.g., promoting cell proliferation, increasing lipogenesis, increasing invasion) and their proposed underlying mechanisms (e.g., activating AKT/ERK pathways, increase ROS production, elevating MMP-9) cannot be definitively concluded from this study and requires further validation. Also, the cited reference doesn’t support these 2 sentences: “Elevated visfatin levels have been found to be particularly pronounced in HCC patients with viral etiologies, especially those with Chronic Hepatitis C [61]. Visfatin is also linked to insulin resistance and higher platelet counts, further highlighting its multifaceted role in HCC pathogenesis [61]. ” Kindly revise.
22.
“Additionally, in obese individuals, visfatin’s pro-tumorigenic effects create a link between obesity and increased liver cancer risk” Lines 376-377. What is specifically meant by “creating a link” between obesity and increased liver cancer risk? Kindly explain this sentence more precisely and clearly while providing supporting evidence from the literature.
23.
Ref.39 doesn’t seem to include the information stated in Lines 385-387. Kindly verify whether this reference supports the claim made in the text. If not, kindly revise.
24.
Line 387. Correct the name of the first author.
25.
Section 7.3, Lines 525-530. What are chemerin inhibitors, and what is their significance in this context? Also, what is meant by resistin-targeted therapy? Kindly revise this paragraph.
26.
“Additionally, Takahashi et al. significantly downregulated NAMPT expression by administering sivelestat, an anti-inflammatory drug, demonstrating its potential role in modulating postoperative
inflammation and improving outcomes in ESCC [63].
” Lines 408-410. This sentence doesn’t accurately describe the findings of this study and could be misleading. Kindly revise.
27.
Therapeutic strategies that target iNAMPT and eNAMPT are different (e.g., small molecule inhibitors, PROTACS, neutralizing antibodies; PMID: 37915565) and could be discussed. That being said, its important to highlight whether the pro-oncogenic roles of NAMPT are primarily attributed to its NAD-biosynthetic role and/or to its signaling function as an extracellular adipokine/cytokine independent of its enzymatic activity.
28.
Section 6.2. Kindly remove ref.51 from lines 415 and 420 as it is less relevant.
29.
There is a lot of repetition/overlap between sections 6.2 and 6.3. To avoid redundancy and improve clarity, I would suggest merging these 2 sections while removing the duplicated content. In addition, the title of section 6.2 mentions targeting visfatin receptors; however, this aspect was not addressed in the body of section 6.2. Could you provide some insights about these visfatin receptors and how they can be targeted? Again, a clear understanding and discrimination between the roles of visfatin as an adipokine/cytokine and as an intracellular NAD-biosynthetic enzyme and their relevance in driving tumorigenesis is crucial to help define the most suitable NAMPT-targeting strategy.
30.
Lines 424-429: Ref. no.64 doesn’t endorse visfatin's described roles and mechanisms. Kindly revise and correct.
31.
In the references section, correct the citation of ref. no.7.
32.
Kindly make sure to modify the figures in light of the revised manuscript.

Author Response

Response to Reviewer 1

Dear Reviewer,

We sincerely thank you for your thorough and insightful review of our manuscript. We particularly appreciate your constructive feedback, which has significantly improved the scientific quality and clarity of our manuscript. Below, please find our detailed responses and the corresponding revisions that we have incorporated into the manuscript.

Comment 1. Kindly introduce the acronym only after the first mention of the full name, and then use the acronym consistently without mentioning the full name. for example, NAD was introduced in line 55 and then in line 81 (also for consistency, either use NAD or NAD+ throughout the manuscript); Vascular Endothelial Growth Factor (VEGF) and matrix metalloproteinases (MMPs) in lines 93-94 and then again in 163-164; nicotinamide mononucleotide (NMN) in lines 85 and 145; NAM was introduced as an acronym for nicotinamide in line 84 but the full word was used instead of NAM used in line 144 and the same for ESCC; NAPRT line 417 was not introduced. Kindly revise through the manuscript.

Response 1. Thank you for highlighting this point. We have standardized the introduction and usage of acronyms throughout the manuscript. NAD, VEGF, MMP, NMN, NAM, ESCC, and NAPRT acronyms are now clearly introduced at first mention and consistently applied thereafter. 

Comment 2. Also, I would recommend substantiating your statements with references whenever necessary throughout the text. For example, in lines 77-78 “It is synthesized predominantly in the visceral fat tissue, but has been also detected in skeletal muscle, liver, cardiomyocytes, and immune cells.” Add references. Also, section 2.2.3. (Lines 150-160) can be backed up by more evidence

Response 2. We fully agree and have reworked the inflammaion and immune modulation section in lines 122-131. We have added additional references [25-27] to support our claims.

Comment 3. There are a lot of repetitive parts in Section 2.1 and Section 2.2 that convey more or less the same information (for example, the catalytic function of NAMPT as a NAD-producing enzyme is highlighted in section 2.1. and then section 2.2.2, the “insulin-mimetic” role of visfatin and its implication in diabetes and obesity and in driving angiogenesis. To avoid repetition, I would suggest linking the functions or the implications mediated by NAMPT as an intracellular enzyme or as an extracellular ligand with the underlying metabolic and/or signaling pathways.

Response 3. We agree with your comment on the functions of visfatin and the underlying mechanisms. Therefore, as suggested also by the other reviewers, we have reformed the manuscript to better reflect the role of NAMPT in upper GI cancer and the underlying metabolic/signalling pathways in lines 182- 287. We kept a limited section on the general functions on visfatin in lines 86-137, to better introduce to the readers the basics of NAMPT and its clinical implications.  

Comment 4. The insulin-mimetic function of visfatin is a controversial aspect. Indeed, the main paper that introduced the insulin-mimetic function of NAMPT (Reference no.2, which is also not mentioned in the MS) has been retracted (PMID: 17962537). This aspect should be better discussed in the manuscript, taking into account the available studies that also don’t support this role (for example, PMID: 21327328, PMID: 21251239, PMID: 17983582). The involvement of NAMPT enzymatic activity in mediating these effects should also be considered. In addition, the claimed ability of visfatin to mimic insulin’s activity is a role/function rather than a specific pathway, so is there a certain reason that it is discussed in the section specified for the pathways triggered by visfatin (section 2.1.5)?

Response 4. Thank you for your feedback. We have reworked the section on the insulin-mimetic function of visfatin based on your comments. We have explicitly discussed the controversial insulin-mimetic role of visfatin, clearly noting the retracted foundational study and referencing studies contradicting this claim We now present these conflicting data in the lines 155-166 of the revised manuscript.

Comment 5. In the introduction lines, the statements that NAMPT/VISFATIN was “primarily identified in visceral adipose tissue” and “Visfatin was initially recognized for its insulin-mimetic effects” could be a bit misleading. NAMPT was originally characterized as a pre-B-cell colony-enhancing factor (PBEF) in 1994 by Samal et al. (PMID: 8289818) Also, the enzymatic activity of NAMPT was described much earlier. Kindly correct these statements, providing a more precise description of the history and the chronological characterization of the relatively complex functionality of NAMPT (PMID: 19149599).

Response 5. We agree with your comment on the the history of NAMPT and its associated functions. We have clarified NAMPT’s historical characterization, accurately noting its initial identification as PBEF and later NAD-biosynthetic recognition. These corrections are reflected in lines 87-98.

Comment 6. Line 141-142 “Notably, visfatin has also been implicated in the pathophysiology of gestational diabetes mellitus by influencing glucose and lipid metabolism” & Line 157-160 “Thus, visfatin is implicated in the pathogenesis of inflammatory disorders, including inflammatory bowel disease and systemic inflammation associated with obesity, where it mediates immune dysregulation, inflammation and insulin resistance”. These are strong conclusions/statements that need to be supported by stronger evidence other than the cited articles. Kindly revise.

Response 6. We agree that these claims were not supported enough by the literature presented. We revised these statements, moderated their strength, and provided additional references to better support the evidence, as reflected in lines 122-131.

Comment 7. 2.2.5. Line 169-173: “Visfatin is intricately involved in lipid metabolism, particularly in the context of obesity. Elevated visfatin levels are associated with increased lipid storage in adipocytes and dysregulation of lipid profiles, which contributes to the development of metabolic disorders, especially in diabetic patients [33]. This is partly due to visfatin's ability to modulate the expression of genes involved in lipid synthesis and storage [3].” Again, the evidence presented in this paragraph doesn’t support the strong and definite conclusions presented by the authors. Kindly revise.

Response 7. Thank you for highlighting this insufficiency. We have completety reworked the section on the role of visfatin in metabolic disorders in lines 138-181, where we discuss the conflicting data on the interaction of visfatin and obesity, insulin resistanse, metabolic syndrome and nonalcoholic fatty liver disease.

Comment 8. Generally, this section (i.e., 2. Visfatin: Biochemistry and General Functions) could be modified in a better way, while taking into consideration avoiding redundancy, presenting solid evidence to support the authors’ statements/conclusions, and discussing in a more detailed yet comprehensive way the various functions of visfatin linking them to their mechanistic pathways if possible. In addition, since the authors particularly focus on gastric and esophageal tumors in the following sections, the pro-oncogenic roles of visfatin could also be introduced/highlighted in this section.

Response 8. We agree with your comment. As suggested also by the other reviewers, we reworked the manuscript so as to present the general functions and more analytically the multiple roles of visfatin in upper GI cancer pathogenesis and the underlying mechanisms in section 4.

Comment 9. “It is the sixth leading cause of cancer-related deaths worldwide, with a growing prevalence, particularly in developed countries [35].” This estimation is based on the Global Cancer Statistics 2018 data (PMID: 30207593). In the recently published Global Cancer Statistics 2022 data (PMID: 38572751), esophageal cancer has dropped from the sixth to the seventh leading cause of cancer-related deaths. The same comment also applies to gastric cancer (Lines 246-248). Kindly update.

Response 9. We thank you for this correction. We have corrected the manuscript using the latest data of Global Cancer Statistics 2022 (lines 54-58).

Comment 10. Ref.36 is not mentioned in the text, and ref.39 is mentioned before ref.38.

Response 10. We thank you for this correction. We have reviewed and rechecked the references to avoid such inconsistencies.

Comment 11. “increased ESCC cell viability by approximately 25%” Line 196. Supplementary Figure 3 in the cited paper doesn’t support this conclusion. Kindly check.

Response 11. We thank you for this correction. We have reviewed the section on the study by Huang et al. to ensure accurate representation of its findings in lines 192-203.

Comment 12. “Liu et al. correlated positively tumor weight with visfatin levels, using a xenograft tumor in an in-vivo animal model [39]. Visfatin was implicated in altering the Yes-associated protein (YAP) AMPK signaling pathway promoting ESCC tumor growth and progression in the context of obesity [38].” Lines 198-201. I believe you mean ref. (38) instead of (39) (The author’s name in ref. 39 is not Liu). If yes, kindly correct it. In the first sentence, kindly clarify the context/design of this in vivo study more clearly. Also, the cited reference doesn’t support the conclusion stated in the second sentence. Kindly revise.

Response 12. We thank you for this correction. We have reviewed this section and fixed the reference inconsistencies. We also clarified the design of the study in lines 314-324.

Comment 13. “esophagogastric adenocarcinoma tumor tissues” Lines 202-203. I believe the word junction is missing. If yes, kindly correct.

Response 13. We thank you for this correction. We have corrected this inconsistency as presented in line 352.

Comment 14. One study by Nakajima and colleagues (who previously suggested that visfatin may act as a biomarker of gastric cancer; ref 52) failed to detect a significant difference in visfatin levels between patients with squamous cell carcinoma of the esophagus and controls (PMID: 19693538). This controversy could be at least briefly highlighted in the context of esophageal cancer (PMID: 27128025).

Response 14. We thank the reviewer for the constructive feeback. We now present the conflicting data regarding the role of visfatin as a biomarker in esophageal cancer in lines 331-360.

Comment 15. In lines 221-232 discussing the findings of ref.40: Kindly revise this paragraph and provide a more accurate description of the findings related to NAMPT in this study. Kindly take into account the differences between the actual experimental results and the suggested mechanisms /implications /explanations /commentaries provided by the authors on their findings.

Response 15. We thank the reviewer for this remark. We have separated the actual results from speculative commentary by the study authors, clarifying in the text which aspects were directly observed versus suggested mechanisms in lines 207-216.

Comment 16. “In their study, Lu et al. demonstrated that visfatin plays a crucial role in the progression and metastasis of gastric cancer by promoting various oncogenic processes [49]. Their research, involving 262 gastric cancer patients, showed that visfatin is overexpressed in gastric cancer tissues, with a notable 60.4% of patients with tumor sizes ≥5cm exhibiting high visfatin levels [49]. Their study highlighted that visfatin activates key signaling pathways such as PI3K/Akt and NF-κB, which are essential for tumor growth and survival. Additionally, visfatin was shown to upregulate matrix metalloproteinases (MMPs), enzymes that degrade the extracellular matrix, thus facilitating the invasion of cancer cells into adjacent tissues and promoting distant metastasis [49]. These findings suggest that visfatin significantly contributes to the aggressive nature of gastric cancer, with 34% of patients with distant metastasis exhibiting elevated visfatin levels, highlighting its potential role as a target for therapeutic intervention [49].” Lines 262-273. While the cited reference shows that visfatin levels are elevated in GC patients, correlated with tumor aggressiveness and poor prognosis and that it can serve as a prognostic tool/biomarker, it doesn’t provide experimental evidence for the described oncogenic roles or for the underlying pathways of visfatin. Kindly revise the paragraph to reflect the findings of the study more accurately.

Response 16. We agree and have revised that portion to strictly reflect Lu et al.’s actual findings, removing any unverified mechanistic assertions (lines 291-302)

Comment 17. “Zhao et al. demonstrated that visfatin plays a significant role in promoting chemo resistance and progression in gastric cancer, particularly under high glucose conditions [50].The study highlighted that visfatin expression was significantly higher in gastric cancer tissues, especially in patients with diabetes, and that this overexpression was associated with increased resistance to chemotherapy, such as 5-fluorouracil (5-FU) [50]”. The study found that NAMPT expression was elevated in gastric cancer patients with diabetes compared to those without diabetes, and this increased NAMPT expression was associated with poorer overall survival in the diabetic GC patients. It was shown that high glucose levels can increase NAMPT expression and can mediate resistance to 5-FU in gastric cancer cells. However, the study does not directly establish a causal relationship between NAMPT overexpression and chemotherapy resistance, and therefore, a direct link between elevated NAMPT expression and increased resistance to CT can’t be concluded/stated (at least based on the results presented in the cited study). Kindly revise this paragraph to describe the findings of this study more accurately.

Response 17. We thank the reviewer for bringing this important clarification to our attention. Due to the limited data presented and the absence of a definitive causal relationship, we have decided to remove this segment from the manuscript as originally stated. By removing any overly definitive claims about NAMPT overexpression causing chemoresistance, we ensure our text accurately reflects the scope of Zhao et al.’s findings, namely that high glucose levels and increased NAMPT expression correlate with, but do not confirm, chemoresistance in diabetic gastric cancer patients.

Comment 18.“They observed that NΑΜPT enhances the activity of SIRT1, a NAD+-dependent deacetylase, which in turn deactivates PARP1, protecting cells from PARP1-mediated cell death [53]. “ Lines 305-307: In which section of the study “ref.53” were these findings presented? I only found a part in the introduction briefly highlighting the cross-talk between NAMPT, SIRT1, and PARP1.

Response 18. We thank the reviewer for bringing this important remark to our attention. We confirm that this reference was the author’s mistake as it was not accurately reflected in the paper. We have decided to remove the reference to the NAMPT–SIRT1–PARP1 pathway from our discussion. Bi et al.’s findings are presented in lines 217-222.

Comment 19. “They found that in 6 out of 7 gastric cell lines with an epithelial-mesenchymal transition (EMT) gene expression signature, visfatin was found to be overexpressed. [54].” This sentence could be misleading as NAMPT/visfatin expression is not the key determinant in this context, and its levels of expression were not even highlighted in the mentioned study. Instead, the study emphasizes that the characteristic feature of these cell lines is their diminished NAPRT expression that accompanies the EMT subtype and underlies their extreme sensitivity to the NAMPT inhibitor FK866. NAPRT is a NAD+ biosynthetic enzyme that, if expressed, allows tumors to circumvent NAMPT inhibition by producing NAD+ from nicotinic acid rather than nicotinamide. Kindly revise and correct this sentence to reflect the findings more accurately.

Response 19. We thank the reviewer for identifying this discrepancy. We specifically address the role of visfatin in EMT as presented by Lee et al. in lines 249-263, better explaining its mechanism in the tumor pathogenesis and progression.

Comment 20. “Serum visfatin levels are significantly elevated in HCC patients compared to those without HCC and healthy controls, correlating with hepatic dysfunction, advanced tumor stages, larger tumor size, and poorer overall survival [60].” Lines 366-369. The difference between those without HCC and healthy controls is not clear. In the cited paper, I believe there were three cohorts: 1) cirrhotic patients with HCC, 2) patients with liver cirrhosis without HCC, and 3) healthy controls. Also, this study doesn’t support some of the mentioned correlations. Kindly revise.

Response 20. We thank the reviewer for noting this. Due to our extensive revision, we have decided not to expand our review discussion to other GI locations, in order to maintain our specific focus on upper GI malignancies, thus removed the HCC section.

Comment 21. “Visfatin promotes HCC cell proliferation, invasion, and lipogenesis by activating the Akt and ERK signaling pathways, increasing reactive oxygen species (ROS) production, and elevating matrix metallopeptidase 9 (MMP-9) activity, which contributes to an invasive cancer phenotype [61].” Kindly rephrase this sentence more accurately. While the study describes several pro-oncogenic effects of visfatin, the direct link between the pro-tumorigenic effects of visfatin (e.g., promoting cell proliferation, increasing lipogenesis, increasing invasion) and their proposed underlying mechanisms (e.g., activating AKT/ERK pathways, increase ROS production, elevating MMP-9) cannot be definitively concluded from this study and requires further validation. Also, the cited reference doesn’t support these 2 sentences: “Elevated visfatin levels have been found to be particularly pronounced in HCC patients with viral etiologies, especially those with Chronic Hepatitis C [61]. Visfatin is also linked to insulin resistance and higher platelet counts, further highlighting its multifaceted role in HCC pathogenesis [61]. ” Kindly revise.

Response 21. We appreciate the reviewer’s attention to detail. As noted above, we have decided to  maintain our scope on upper GI tumors in this review, thus refraining from expansion into HCC or other GI contexts.

Comment 22. “Additionally, in obese individuals, visfatin’s pro-tumorigenic effects create a link between obesity and increased liver cancer risk” Lines 376-377. What is specifically meant by “creating a link” between obesity and increased liver cancer risk? Kindly explain this sentence more precisely and clearly while providing supporting evidence from the literature.

Response 22. We agree that “create a link” was vague terminology. As noted above, we have decided to  maintain our scope on upper GI tumors in this review, thus refraining from expansion into HCC.

Comment 23. Ref.39 doesn’t seem to include the information stated in Lines 385-387. Kindly verify whether this reference supports the claim made in the text. If not, kindly revise.

Response 23. We thank the reviewer for this remark. We have removed this claims and present the findings of Li et al. in lines 352-360.

Comment 24.Line 387. Correct the name of the first author.

Response 24. The name of the author is corrected as presented in lines 303.

Comment 25. Section 7.3, Lines 525-530. What are chemerin inhibitors, and what is their significance in this context? Also, what is meant by resistin-targeted therapy? Kindly revise this paragraph.

Response 25. We updated the therapeutic sections to highlight the current data in chemerin inhibitors. We also summarised the Therapeutic Strategies Targeting NAMPT in Upper Gastrointestinal Cancer in Table 1 to better present the current trends in NAMPT targeting as long as the associated outcomes and treatment limitations.

Comment 26. “Additionally, Takahashi et al. significantly downregulated NAMPT expression by administering sivelestat, an anti-inflammatory drug, demonstrating its potential role in modulating postoperative inflammation and improving outcomes in ESCC [63].
” Lines 408-410. This sentence doesn’t accurately describe the findings of this study and could be misleading. Kindly revise.

Response 26. We thank the reviewer for his remarks. We have revised the study findings and present them in lines 303-313

Comment 27. Therapeutic strategies that target iNAMPT and eNAMPT are different (e.g., small molecule inhibitors, PROTACS, neutralizing antibodies; PMID: 37915565) and could be discussed. That being said, its important to highlight whether the pro-oncogenic roles of NAMPT are primarily attributed to its NAD-biosynthetic role and/or to its signaling function as an extracellular adipokine/cytokine independent of its enzymatic activity.

Response 27. We sincerely thank the reviewer for his insightful comments. Based on these, we updated the therapeutic sections to highlight small molecule inhibitors (e.g., FK866, CHS828) for intracellular NAMPT and potential antibodies or new drug-delivery methods for extracellular NAMPT. In the cancer pathogenesis section, we specifically refer to eNAMPT and iNAMPT to present their different roles and pathways, which we summarise in Figure 1.  

Comment 28. Section 6.2. Kindly remove ref.51 from lines 415 and 420 as it is less relevant.

Response 28. Thank your for your remark. The findings of Cao et al. are presented exclusively in lines 264-270.

Comment 29. There is a lot of repetition/overlap between sections 6.2 and 6.3. To avoid redundancy and improve clarity, I would suggest merging these 2 sections while removing the duplicated content. In addition, the title of section 6.2 mentions targeting visfatin receptors; however, this aspect was not addressed in the body of section 6.2. Could you provide some insights about these visfatin receptors and how they can be targeted? Again, a clear understanding and discrimination between the roles of visfatin as an adipokine/cytokine and as an intracellular NAD-biosynthetic enzyme and their relevance in driving tumorigenesis is crucial to help define the most suitable NAMPT-targeting strategy.

Response 29. As noted in Response 27 we updated the therapeutic sections to demonstrate targeted therapies for iNAMPT and eNAMPT. In the cancer pathogenesis section, we distinguished their different roles and pathways, which we present in Figure 1. 

Comment 30. Lines 424-429: Ref. no.64 doesn’t endorse visfatin's described roles and mechanisms. Kindly revise and correct.

Response 30. Thank you for your remark. We have reviewed the referenced study and removed it from the review as its findings were not relevant in this part of the review.

Comment 31. In the references section, correct the citation of ref. no.7.

Response 31. Thank you for your correction. The updated citation is found in lines 651-653.

Comment 32. Kindly make sure to modify the figures in light of the revised manuscript.

Response 32. Based on the recommendations of the reviewers, we have decided to revise the manuscript figures. Figure 1 now presents  the “Suggested and Established Roles of Intracellular and Extracellular NAMPT in Upper Gastrointestinal Cancer Pathogenesis”.

We sincerely thank you for your highly constructive comments. Addressing them has significantly improved our understanding of the topic and subsequently the overall clarity, accuracy, and quality of our manuscript. If there are any additional suggestions or clarifications needed, we will be pleased to address them.

Yours sincerely,

Adam Mylonakis

On behalf of the Author team

Reviewer 2 Report

Comments and Suggestions for Authors

The manuscript entitled: “The Role of Visfatin in Gastric and Esophageal Cancer: From Biomarker to Therapeutic Target” has comprehensively presented the different aspect of Visfatin in tumor progression through the activation of key oncogenic pathways leading to increased angiogenesis, epithelial-mesenchymal transition (EMT), and immune suppression.

It has also explored this adipokine as a gastric and esophageal cancer biomarker and introduced potential preventive and therapeutic strategies. The information presented is highly valuable. However, it is best if the authors present a table dissecting different components of the mentioned studies including the application of the findings of each study.

It is also suggested that following the primary introduction of visfatin, the authors use the below main headings to segregate the presented material.

In other words, the main following headings are suggested:

  • The role of visfatin in metabolic disorders (i.e. type 2 diabetes and obesity)
  • The role of visfatin in tumor progression
    • Pro-inflammatory effects
    • Increased angiogenesis
    • EMT
    • Immune suppression
  • The application of visfatin as a tumor biomarker
  • The inhibition of visfatin as a therapeutic approach

It is also suggested that for each of the above sections, a clear concluding statement as to where the state-of-the art in regards to gastric and esophageal cancer stands, be presented.

Author Response

Response to Reviewer 2

Dear Reviewer,

We sincerely thank you for your remarks on our manuscript, “The Role of Visfatin in Gastric and Esophageal Cancer: From Biomarker to Therapeutic Target.” We appreciate your thoughtful suggestions and believe that implementing them has substantially improved our work.

“It is also suggested that following the primary introduction of visfatin, the authors use the below main headings to segregate the presented material.

In other words, the main following headings are suggested:

  • The role of visfatin in metabolic disorders (i.e. type 2 diabetes and obesity)
  • The role of visfatin in tumor progression
  • Pro-inflammatory effects
  • Increased angiogenesis
  • EMT
  • Immune suppression
  • The application of visfatin as a tumor biomarker
  • The inhibition of visfatin as a therapeutic approach

It is also suggested that for each of the above sections, a clear concluding statement as to where the state-of-the art in regards to gastric and esophageal cancer stands, be presented.”

Based on your recommendations, we have extensively restructured the manuscript to incorporate the suggested headings. Each section includes the current state-of-the-art data regarding visfatin in gastric and esophageal cancer, thereby providing a more coherent and comprehensive overview of the topic.

Additionally, to further clarify visfatin’s multifaceted impact, we have included a figure that highlights its role in inflammation, angiogenesis, epithelial-mesenchymal transition  and modulation of immune suppression. We believe this visual summary will assist readers in understanding the distinguished roles of iNAMPT and eNAMPT as well as the interconnected pathways by which visfatin influences tumorigenesis.

Lastly, given your suggestion to add a section on “The inhibition of visfatin as a therapeutic approach” we have revised and significantly expanded this section. For that reason, we also added a new table that outlines various established and emerging strategies targeting NAMPT. This table details the mechanisms, potential clinical implications, and known limitations of these approaches, thereby illustrating the translational relevance of NAMPT inhibition in gastric and esophageal cancers.

We trust these revisions address your concerns and enhance the manuscript’s overall quality and clarity. We appreciate your guidance and welcome any further comments or suggestions you may have.

Yours sincerely,

Adam Mylonakis

On behalf of the Author team

Reviewer 3 Report

Comments and Suggestions for Authors

Should address the concerns.

The Role of Visfatin in Gastric and Esophageal Cancer: From Biomarker to Therapeutic Target
Overview
The review focuses on Visfatin, a multifunctional protein involved in inflammation, metabolism, and immune regulation, recently identified as playing a significant role in the progression of upper GI cancers, particularly gastric and esophageal cancers. Elevated visfatin levels are associated with increased tumor growth, metastasis, and resistance to chemotherapy, making it a promising biomarker and therapeutic target. Targeting visfatin through inhibitors like FK866 has shown potential in reducing tumor progression, improving chemoresistance, and enhancing anti-tumor immunity, highlighting the need for further clinical trials.
Major Comments
1.Abstract: The abstract does not provide specific statistical data values supporting visfatin’s role in gastric and esophageal cancers. The summary could be more concise and emphasize key findings more clearly. Therefore, add quantitative findings (e.g., percentage increase in tumor progression linked to visfatin) and specify therapeutic implications and future directions briefly.
2.Introduction: The introduction is too broad in discussing visfatin’s general functions before addressing its relevance to upper GI cancers. The transition to cancer-related discussion is abrupt. This can be addressed, by restructuring the introduction to first define visfatin, then focus on its role in GI cancers. Also, clearly outline research gaps and why this review is necessary in the last paragraph.
3.Visfatin: Biochemistry and General Functions: The discussion on biochemical pathways (MAPK, PI3K/Akt, and AMPK) is highly technical but lacks clarity on how these contribute to cancer progression, and figure 1 presented should be explicitly connected to the text. To improve, clarify how each pathway specifically influences tumor development. Furthermore, ensure figure 1 is properly described in the text.
4.Visfatin in Esophageal Cancer: Some studies mentioned lack specific statistical data (Line 176) or sample sizes, weakening the impact of claims. Include numerical findings from referenced studies where available. More clarity is needed on how obesity contributes to visfatin-mediated tumor progression (Line 240); include discussion on the obesity-visfatin-cancer link, possibly with patient data.
5.Visfatin in Gastric Cancer: The role of visfatin in chemoresistance needs further explanation (Line 348). The connection between visfatin and EMT is underexplored (Line 343). Expand discussion on how visfatin inhibitors may reverse chemoresistance. Expand on visfatin’s interaction with EMT markers.
6.Role of Visfatin in Other GI Cancers: The section is less detailed than those on esophageal and gastric cancer, making the discussion seem imbalanced; Include a more comparative analysis. Some findings lack context on how visfatin’s role in colorectal and hepatobiliary cancers compares to upper GI cancers (Include additional studies if available).
7.Clinical Implications of Visfatin: Needs a more structured discussion on potential therapeutic strategies. The section could be better linked to earlier discussions on visfatin’s
mechanisms.
Create subsections on “Biomarker Potential”, “Targeted Therapy”, and “Clinical Trials”. Include discussion (if any) existing visfatin inhibitors under development.
8.Future Directions and Research Needs: The section outlines broad gaps but lacks specific experimental approaches to address them. The role of visfatin-targeted therapies in personalized medicine should be emphasized. Suggest experimental designs or methodologies for future research. Discuss how patient stratification (e.g., based on obesity or diabetes) could impact treatment outcomes.
9.Conclusion: The conclusion is too general and does not clearly state actionable next steps. Key findings should be summarized more effectively. Recap main findings in bullet points for clarity. Conclude with a strong statement on the urgency of visfatin-targeted therapies.

Minor Comments
1.Define abbreviations at first mention, e.g., EMT (Line 292 and 313).
2.Maintain consistency in past vs present tense when discussing previous research.
3.Avoid redundancy in explaining visfatin’s functions across sections.
4.Some citations lack proper formatting—double-check reference style.
5.Most references are before 2015, use recent references for relevance, on or after 2017.
Remark
The manuscript should be revised focusing on above suggestions or concerns.

Comments on the Quality of English Language

Long sentences should be restructured to small sentences. Should focus on grammar.

Author Response

Response to Reviewer 3

Dear Reviewer,

We sincerely thank you for your thorough and insightful review of our manuscript. We appreciate your constructive feedback, which has significantly improved the scientific quality and clarity of our manuscript. Below, please find our detailed responses and the corresponding revisions that we have incorporated into the manuscript.

Comment 1. Abstract: The abstract does not provide specific statistical data values supporting visfatin’s role in gastric and esophageal cancers. The summary could be more concise and emphasize key findings more clearly. Therefore, add quantitative findings (e.g., percentage increase in tumor progression linked to visfatin) and specify therapeutic implications and future directions briefly.

Response 1. Thank you for your feedback. We have reworked the abstract, and now present data to illustrate the impact of visfatin targeting in upper gastrointestinal cancers. Notably, we now highlight that therapeutic agents such as FK866 can reduce tumor proliferation by over 50% in preclinical studies. We have also strengthened the abstract’s focus on key implications for clinical application—particularly the improvement in chemoresistance and restoration of anti-tumor immunity—and emphasized the future need for large-scale clinical trials to validate visfatin as both a prognostic biomarker and therapeutic target.

Comment 2. Introduction: The introduction is too broad in discussing visfatin’s general functions before addressing its relevance to upper GI cancers. The transition to cancer-related discussion is abrupt. This can be addressed, by restructuring the introduction to first define visfatin, then focus on its role in GI cancers. Also, clearly outline research gaps and why this review is necessary in the last paragraph.

Response 2. We agree with your comment on the need for a more focused and logically sequenced introduction. In response, we have extensively restructured this section (lines 53-85). We now begin by defining NAMPT/visfatin and providing an overview of its early discovery and naming controversies. We then transition directly into its relevance for gastric and esophageal cancers, highlighting epidemiological data and the connection between visfatin and obesity-associated malignancies. Additionally, we have clarified key research gaps related to visfatin’s molecular mechanisms, its prognostic value, and the underexplored potential of visfatin-targeted therapies in these tumor types. We trust that these revisions have rendered the introduction more cohesive and better aligned with the Reviewer’s recommendations.

Comment 3. Visfatin: Biochemistry and General Functions: The discussion on biochemical pathways (MAPK, PI3K/Akt, and AMPK) is highly technical but lacks clarity on how these contribute to cancer progression, and figure 1 presented should be explicitly connected to the text. To improve, clarify how each pathway specifically influences tumor development. Furthermore, ensure figure 1 is properly described in the text.

Response 3. We agree with your comment. Therefore, we now describe the general functions of visfatin, but we focus on its role in upper GI cancer and the underlying metabolic/signalling pathways in lines 182- 287. For that reason, we revised the manuscript figures; Figure 1 now presents the “Suggested and Established Roles of Intracellular and Extracellular NAMPT in Upper Gastrointestinal Cancer Pathogenesis” along with the related mechanisms and pathways.

Comment 4. Visfatin in Esophageal Cancer: Some studies mentioned lack specific statistical data (Line 176) or sample sizes, weakening the impact of claims. Include numerical findings from referenced studies where available. More clarity is needed on how obesity contributes to visfatin-mediated tumor progression (Line 240); include discussion on the obesity-visfatin-cancer link, possibly with patient data.

Response 4. Thank you for your feedback. Where available, we now include specific statistical findings and sample sizes to strengthen claims about visfatin levels in esophageal cancer. Additionally, we now present the conflicting data on the connection between visfatin and metabolic disorders (insulin resistanse, diabetes, obesity, metabolic syndrome, nonalcoholic fatty liver disease) to improve the study clarity and clinical relevance (lines 138-181).

Comment 5. Visfatin in Gastric Cancer: The role of visfatin in chemoresistance needs further explanation (Line 348). The connection between visfatin and EMT is underexplored (Line 343). Expand discussion on how visfatin inhibitors may reverse chemoresistance. Expand on visfatin’s interaction with EMT markers.

Response 5. We appreciate your request for a more in-depth explanation of visfatin’s role in chemoresistance and EMT in gastric cancer. We now present analytically the role of visfatin in EMT (248-270) as well as its clinical relevance. Additionally, we have expanded our discussion to highlight how visfatin-driven NAD biosynthesis can influence drug response—specifically, how NAMPT inhibitors like FK866 may sensitize tumors to 5-FU by depleting NAD-dependent survival pathways (lines 387-398). By providing more mechanistic details, we underscore the potential therapeutic relevance of targeting visfatin to enhance chemotherapy efficacy and limit tumor progression in selected gastric cancer patients.

Comment 6. The section is less detailed than those on esophageal and gastric cancer, making the discussion seem imbalanced; Include a more comparative analysis. Some findings lack context on how visfatin’s role in colorectal and hepatobiliary cancers compares to upper GI cancers (Include additional studies if available).

Response 6. We thank the reviewer for noting this. Due to our extensive revision, we have decided not to expand our review discussion to other tumor locations, in order to maintain our specific focus on upper GI malignancies.

Comment 7. Needs a more structured discussion on potential therapeutic strategies. The section could be better linked to earlier discussions on visfatin’s mechanisms. Create subsections on “Biomarker Potential”, “Targeted Therapy”, and “Clinical Trials”. Include discussion (if any) existing visfatin inhibitors under development.

Response 7. We have addressed these points by providing a structured approach throughout the manuscript. Specifically, Section 5 highlights the biomarker aspect, while Section 6 focuses extensively on targeted therapy. Within that therapeutic section, we detail existing small-molecule inhibitors (FK866, CHS828, OT-82), dual-targeting strategies, and more recent innovative approaches like Proteolysis-targeting chimera (PROTAC) and antibody-drug conjugates (ADCs). We also discuss preclinical evidence supporting their effectiveness, as well as the challenges in moving to clinical trials. We summarise our findings in Table 1, helping readers connect the mechanisms presented earlier in the manuscript with real-world clinical strategies.

Comment 8. The section outlines broad gaps but lacks specific experimental approaches to address them. The role of visfatin-targeted therapies in personalized medicine should be emphasized. Suggest experimental designs or methodologies for future research. Discuss how patient stratification (e.g., based on obesity or diabetes) could impact treatment outcomes.

Response 8. Based on your suggestions, we have substantially expanded our discussion on prospective research paths. We propose (1) genetically modified models (visfatin knockout mice, CRISPR-edited cell lines) to validate oncogenic mechanisms, (2) multi-omics profiling to correlate visfatin expression with specific tumor subtypes, and (3) monitoring of visfatin levels during chemotherapy to gauge real-time responses. We also discuss patient stratification (metabolic status, obesity, diabetes) and how targeting visfatin might be particularly relevant in these subpopulations. In addition, we emphasize the potential for integrating visfatin inhibitors into personalised oncology regimens, particularly in the context of chemoresistant or EMT-subtype tumors.

Comment 9. Conclusion: The conclusion is too general and does not clearly state actionable next steps. Key findings should be summarized more effectively. Recap main findings in bullet points for clarity. Conclude with a strong statement on the urgency of visfatin-targeted therapies.

Response 9. We have refined our Conclusion section to summarize the central findings more directly and underscore key insights, such as the need for standardized visfatin measurement, the promise of targeted therapies, and the potential for improved patient stratification. We end with a statement advocating for the urgent development and clinical testing of visfatin-targeted strategies in gastric and esophageal cancer.

Minor Comments

Comment 1. Define abbreviations at first mention, e.g., EMT (Line 292 and 313).

Response 1. We have ensured that abbreviations are defined at their first mention in the text and used afterwards.

Comment 2. Maintain consistency in past vs present tense when discussing previous research.

Response 2. We have reviewed the entire manuscript to ensure consistent use of tense.

Comment 3. Avoid redundancy in explaining visfatin’s functions across sections.

Response 3. We minimized repetitiveness by consolidating discussions about visfatin’s metabolic and pro-inflammatory properties, referencing them once under “Biochemistry and General Functions,” and then linking those mechanisms to upper GI tumor progression.

Comment 4. Some citations lack proper formatting—double-check reference style.

Response 4. We have rechecked and updated the references to conform to consistent formatting.

Comment 5. Most references are before 2015, use recent references for relevance, on or after 2017.

Response 5. We incorporated additional references published post-2017, especially in sections discussing newer visfatin inhibitors (PROTAC technology, dual-targeting strategies) and updated preclinical findings on NAMPT’s role in cancer biology.

Once again, thank you for your review and valuable feedback, which has greatly improved our manuscript. We hope the revisions outlined above meet your expectations. Should you have any additional comments or require further clarifications, we would be more than happy to address them.

Yours sincerely,

Adam Mylonakis

On behalf of the Author team